# E5-V: Universal Embeddings with Multimodal Large Language Models

## Abstract

Multimodal large language models (MLLMs) have shown promising advancements in general visual and language understanding. However, the representation of multimodal information using MLLMs remains largely unexplored. In this work, we introduce a new framework, E5-V, designed to adapt MLLMs for achieving universal multimodal embeddings. Our findings highlight the significant potential of MLLMs in representing multimodal inputs compared to previous approaches. By leveraging MLLMs with prompts, E5-V effectively bridges the modality gap between different types of inputs, demonstrating strong performance in multimodal embeddings even without fine-tuning. We propose a single modality training approach for E5-V, where the model is trained exclusively on text pairs. This method demonstrates significant improvements over traditional multimodal training on image-text pairs, while reducing training costs by approximately 95%. Additionally, it eliminates the need for costly multimodal training data collection. Extensive experiments across four types of tasks demonstrate the effectiveness of E5-V. As a universal multimodal model, E5-V not only achieves but often surpasses state-of-the-art performance in each task, despite being trained on a single modality.

## 1 Introduction

With the development of MLLMs, there is an increasing need for embedding models to represent multimodal inputs. Although CLIP Radford et al. (2021) shows impressive results in text-image retrieval by aligning visual and language representations with contrastive learning, it struggles to represent interleaved visual and language inputs. Moreover, the text encoder of CLIP demonstrates a low capacity for understanding complicated text Zhang et al. (2024). To achieve universal multimodal representation, some works Wei et al. (2023); Zhou et al. (2024) continue to train CLIP on interleaved image-text data, while collecting such data can be challenging and may require GPT-4 to synthesize data Zhou et al. (2024) or manualy annotated.

Recent works demonstrate Wang et al. (2023); Jiang et al. (2023) that scaling up the size of text embedding models leads to better performance. However, replicating this scaling approach for universal multimodal embeddings poses significant challenges and expenses, which arises from the unstable for scaling CLIP and the complexity of collecting extensive multimodal datasets Sun et al. (2023). Nevertheless, previous works like adapting CLIP to universal multimodal embeddings still has shortcomings, such as poor language understanding, limited real-world knowledge, and shallow fusion of visual and linguistic information.

In this work, we introduce a new framework, called E5-V, to directly adapt MLLMs instead of CLIP like models for achieving universal multimodal embeddings. There are several advantages to representing multimodal information with MLLMs: First, benefiting from interleaved visual and language training, MLLMs can initially learn to represent multimodal information according to their meanings with prompt. Second, MLLMs are capable of representing interleaved visual and language inputs to handle tasks like composed image retrieval. Third, MLLMs have stronger language understanding and reasoning capabilities compared to CLIP.

However, since MLLMs are not initially trained with contrastive learning to represent inputs as embeddings, it can be challenging for them to represent multimodal inputs as well as CLIP, which performs contrastive learning on large-scale text-image pairs. In this work, we propose a prompt-based representation method to adapt MLLMs for multimodal embeddings inspired by Jiang et al.

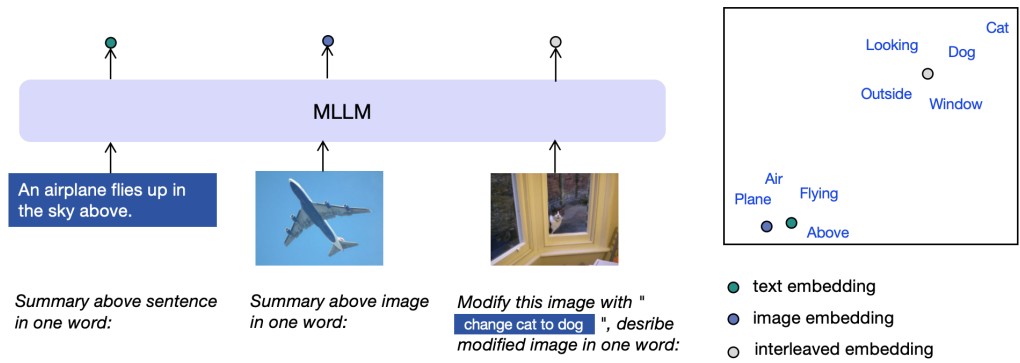

Figure 1: 2D visualization of multimodal embeddings and token embeddings in MLLM. Words correspond to the tokens in MLLM, and dots represent the multimodal embeddings. Our method unifies different multimodal embeddings from MLLM into the same space corresponding to their meanings without fine-tuning.

(2023). By explicitly instructing MLLMs to represent multimodal inputs into words in Figure 1, this method initially unifies multimodal embeddings into the same space, which directly remove the `modality gap` Liang et al. (2022) in multimodal embeddings.

By unifying multimodal embeddings into the same space, MLLMs are able to achieve robust multimodal embedding performance through single modality training with only on text inputs. This eliminates the need for expensive multimodal training data collection. By focusing solely on text data, we can remove other components, such as the visual encoder, in the MLLMs during training and decrease the input size, significantly reducing the training cost. Compared to multimodal training, we observe training solely on text pairs even help MLLMs better represent multimodal inputs than image-text pairs, and find text pairs can be more effective in contrastive learning than image-text pairs.

To validate the effectiveness of E5-V, we conduct experiments on various tasks: text-image retrieval, composed image retrieval, sentence embeddings, and image-image retrieval. By comparing E5-V with the strong baselines of each task, we demonstrate the effectiveness of E5-V in representing multimodal information, which achieves competitive performance on all tasks as a universal multimodal embeddings model trained on text pairs only.

Our contributions are as follows:

- We study how to achieve universal multimodal embeddings by leveraging MLLMs. By designing prompts to project multimodal inputs into the same embedding spaces, we show that MLLMs can represent multimodal inputs correctly even without fine-tuning.

- We introduce a new framework, E5-V, to adapt MLLMs for achieving universal multimodal embeddings. With single modality training on text pairs, E5-v even achieves better multimodal embeddings than image-text pairs.

- Extensive experiments on text-image retrieval and composed image retrieval tasks demonstrate the effectiveness of E5-V in representing multimodal information. E5-V successfully transfers single modality representation capabilities to multimodal embeddings by following task-specific prompts that were not included in the training data.

## 2 RELATED WORK

### 2.1 MULTIMODAL LARGE LANGUAGE MODELS

With the success of LLMs, there is a trend to extend LLMs to handle multimodal information, called MLLMs. MLLMs, such as BLIP Li et al. (2023), KOSMOS Huang et al. (2024), LLaMA-

Adapter Gao et al. (2023), and LLaVA Liu et al. (2024c;b;a), show promising progress in multimodal information understanding and reasoning. To achieve this, a typical MLLM is composed of an LLM, a modality encoder, and a projector to connect them. The modality encoder projects raw multimodal inputs into vectors to connect with LLMs Yin et al. (2023).

One efficient method Gao et al. (2023); Liu et al. (2024c) is to directly use a pretrained LLM and a pretrained modality encoder, such as CLIP Radford et al. (2021). To achieve this, LLaVA uses two training stages. The first stage aligns the text and image with image-text pairs by only training the projector between LLM and modality encoder, and the second stage fine-tunes the model on a visual instruction dataset, which ensure it can follow complex instructions like LLM, such as represent the multimodal inputs in our work.

While the impressive performance of MLLMs in understanding multimodal information and instruction following, the representation of multimodal information using MLLMs remains largely unexplored. Although recent studies Wang et al. (2023); Jiang et al. (2023) have shown advancements in embedding texts with LLMs and scaling up has demonstrated improved performance in text representation, a significant challenge persists: the batch sizes requires to train multimodal embeddings models such as CLIP is signficantly larger than text embeddings models. For example, CLIP requires a batch size of 32k samples with contrastive learning, while the text embedding models such as E5 Wang et al. (2023) only requires a batch size of 2k samples. Limited by the size of MLLMs, it can be very challenging to use the similar batch size like CLIP to train robust multimodal embeddings models.

## 2.2 MULTIMODAL EMBEDDINGS

CLIP Radford et al. (2021), as a pioneering work on multimodal embeddings, has been widely used in subsequent works. CLIP uses separate encoders for image and text by aligning them with contrastive learning on large-scale image-text pairs. Despite its strong performance in text-image retrieval, CLIP has several limitations due to its internal framework. The text encoder of CLIP has a low capacity for understanding complicated text because it is pretrained on short image captions, which also limits CLIP's performance on long text retrieval Zhang et al. (2024). Additionally, due to the use of separate encoders, CLIP struggles to represent interleaved visual and language inputs, such as in composed image retrieval Liu et al. (2021); Wu et al. (2021).

To achieve universal multimodal embeddings, several works, such as UNIIR Wei et al. (2023), fine-tune CLIP with a fusion model to integrate visual and language information. Other works, like VISTA Zhou et al. (2024) or UniVL-DR Liu et al. (2022), feed the text embedding models with CLIP outputs to incorporate visual information. However, this approach can harm the original text-image retrieval performance of CLIP and makes it difficult for the text embedding models to understand visual information using only contrastive learning. As a result, these methods show poor zero-shot performance on composed image retrieval tasks. Moreover, these methods require large interleaved training data to achieve universal multimodal embeddings. Collecting such high-quality interleaved pairs for performing contrastive learning is more challenging than gathering image-text pairs or text pairs. This process can require complex annotation and sometimes even synthesizing data from GPT-4 Zhou et al. (2024).

## 3 E5-V

### 3.1 UNIFYING MULTIMODAL EMBEDDINGS

Previous works Liang et al. (2022) have demonstrated the existence of a `modality gap` between text and image embeddings in multimodal models like CLIP Radford et al. (2021), which can negatively impact the performance of multimodal embeddings. Similarly, we observe this phenomenon when using MLLMs to represent multimodal inputs.

We visualize the distribution of multimodal embeddings from MLLM in Figure 3a following Liang et al. (2022). For implementation, we use the last token embeddings of LLaVA-NeXT-8B Li et al. (2024) to represent the images and captions of COCO. The embeddings are obtained directly from MLLM without fine-tuning and visualized with PCA. Compared to CLIP, although MLLM represents

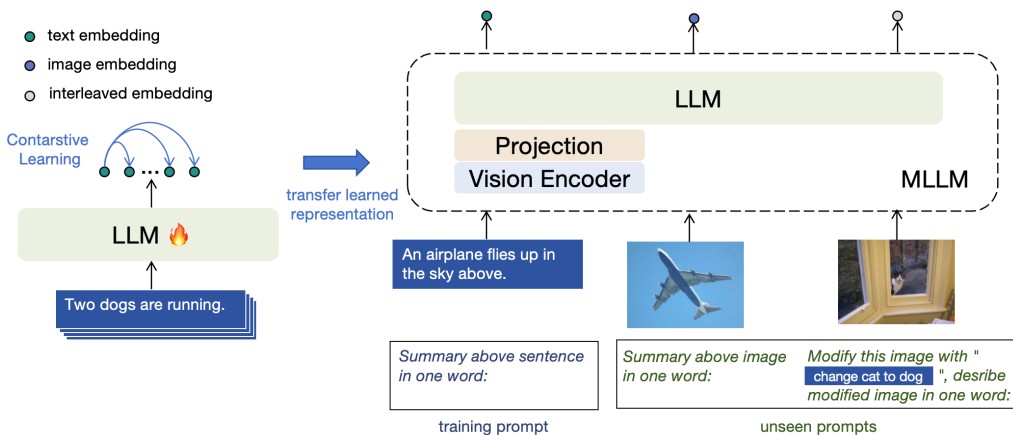

Figure 2: Single modality training in E5-V. By unifying multimodal representations into the same embedding space with prompts, E5-V improves multimodal embeddings using only contrastive learning on text pairs. During training, we remove the modality encoder and projector in MLLM.

the image and text with the same encoder, the multimodal embeddings from MLLM show a clear `modality gap` between text and image embeddings.

To unify multimodal embeddings, we propose a prompt-based representation method with MLLMs inspired by previous text embedding work Jiang et al. (2023). The key idea is to explicitly instruct MLLMs to represent the multimodal inputs into words. We can use prompts like <text> \n *Summary of the above sentence in one word:* to represent the text and <image> \n *Summary above image in one word:* to represent the image. We notice these prompts directly remove the `modality gap` between text and image embeddings, as shown in Figure 3b. For the design of the prompts, it has two parts: the first part is about extracting the meaning of the multimodal inputs, and the second part is about compressing the meaning into the next token

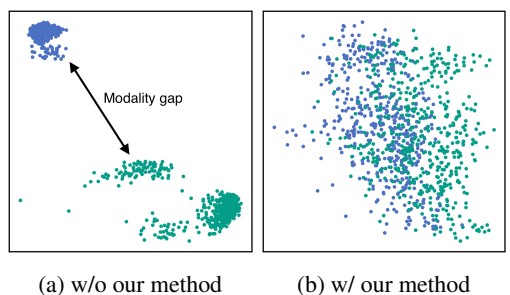

(a) w/o our method      (b) w/ our method

Figure 3: Distribution of image embeddings and text embeddings from MLLM without and with our representation method.

embeddings and unifying the multimodal embeddings by using *in one word:*. Specifically, the embeddings of image and caption about a plane in Figure 1 will have a close distance to the token embeddings of "Plane", "Air", "Flying" and "Above", which represent multimodal inputs based on the corresponding meaning instead of their modality. By removing `modality gap`, it also allows MLLMs to represent interleaved inputs for tasks like composed image retrieval. We demonstrate that our method significantly improves MLLM performance on multimodal retrieval tasks in Table 6.

## 3.2 SINGLE MODALITY TRAINING

By unifying multimodal embeddings, we propose single modality training for multimodal embeddings, as shown in Figure 2. By removing `modality gap` in the embeddings, we can transfer the single modality representation capabilities to multimodal embeddings by training on text pairs only. In this way, our method is trained without any visual or interleaved inputs and no longer relies on multimodal training data, which can be difficult to collect.

To achieve it, E5-V trains MLLMs with contrastive learning on text pairs. Since there are no visual inputs during training, we remove the modality encoder and projector and only remain the LLM of MLLM. For the training data, we simply use sentence pairs from NLI datasets following Gao et al. (2021), which have no relation to the multimodal tasks. Each sentence pair $(x_i, x_i^+, x_i^-)$ has a

positive sentence $x_i^+$ and a negative sentence $x_i^-$ for the input sentence $x_i$. We use the prompt *<text>* \n *Summary above sentence in one word:* to embed the sentence pairs into $(\mathbf{h}_i, \mathbf{h}_i^+, \mathbf{h}_i^-)$. The training objective is following:

$$\mathcal{L} = -\log \frac{e^{\cos(\mathbf{h}_i, \mathbf{h}_i^+)/\tau}}{\sum_{j=1}^N \left( e^{\cos(\mathbf{h}_i, \mathbf{h}_j^+)/\tau} + e^{\cos(\mathbf{h}_i, \mathbf{h}_j^-)/\tau} \right)} \tag{1}$$

where $\tau$ is the temperature hyperparameter and $N$ is the batch size in contrastive learning. Compared to multimodal training, we find that single modality training achieves better performance on multimodal retrieval tasks while significantly reducing training cost, as shown in Table 7.

## 4 EXPERIMENTS

We evaluate E5-V on four tasks: text-image retrieval, composed image retrieval, sentence embeddings, and image-image retrieval to demonstrate the effectiveness of E5-V in representing multimodal information. All tasks are evaluated in a zero-shot setting with the same model without additional fine-tuning on specific datasets.

For the backbone of E5-V, we use LLaVA-NeXT-8B Li et al. (2024), which builds on LLaMA-3 8B Gao et al. (2023), with a frozen CLIP `ViT-L` as the visual encoder. For the training data, we use NLI sentence pairs from Gao et al. (2021), with around 273k sentence pairs. We fine-tune the LLM of LLaVA-NeXT-8B with 1000 steps and 768 batch size. To save the GPU memory, we use QLoRA Dettmers et al. (2024) and gradient checkpointing with DeepSpeed ZeRO-2. For the prompts in training, we use *<text>* \n *Summary above sentence in one word:*, where *<text>* is the placeholder for the input sentence, and use the last token embeddings to represent the embeddings for contrastive learning. We also report the performance with other MLLMs in Appendix C.

### 4.1 TEXT-IMAGE RETRIEVAL

We first benchmark E5-V on text-image retrieval with Flickr30K Young et al. (2014) and COCO Lin et al. (2014) to evaluate zero-shot image retrieval and zero-shot text retrieval performance. For the baselines, we select the following text-image retrieval models: CLIP with `ViT-B` and `ViT-L` Radford et al. (2021), BLIP with `ViT-L` Li et al. (2022), and the large CLIP model EVA-02-CLIP with 5B parameters Sun et al. (2023). All baselines are trained with contrastive learning on large-scale image-text pairs using separate visual and language encoders, while cannot represent interleaved visual and language inputs. For the prompt used in text-image retrieval tasks, we use the following prompts to represent image and text inputs, respectively:

```
Text prompt:
<text>
Summary above sentence in on word:
Image prompt:
<image>
Summary above image in one word:
```

We report Recall@K (R@K) for K=1, 5, 10 with image retrieval and text retrieval in Table 1. Compared to strong baselines, E5-V, as a universal multimodal embeddings model, achieves competitive performance on both the Flickr30K and COCO datasets.

Compared to EVA-02-CLIP, which uses a 4.4B visual encoder with contrastive learning on large-scale image-text pairs Sun et al. (2023), E5-V shows a better ability for zero-shot image retrieval, while it is only trained on text pairs with contrastive learning. It is worth noting that E5-V uses the same visual encoder as CLIP `ViT-L` and keeps it frozen during training. Although E5-V shares the same visual encoder with CLIP, referring to the same way to encode visual inputs, E5-V demonstrates significantly better performance than CLIP on both the Flickr30K and COCO datasets for image retrieval and text retrieval tasks. Specifically, in image retrieval tasks, E5-V outperforms CLIP `ViT-L` by 12.2% on Flickr30K and 15.0% on COCO with Recall@1.

E5-V shows a strong ability to transfer single modality representation capabilities to multimodal embeddings by following task-specific prompts that were not included in the training data. It also

seamlessly integrates visual and language information into the same embedding space with prompts. For unseen prompt in training, E5-V can successfully follow it like "*Summary the above image in one word:*" to represent the image according to its semantics.

| Method | image retrieval | | | | | | text retrieval | | | | | |
|---|---|---|---|---|---|---|---|---|---|---|---|---|
| | Flickr30K | | | COCO | | | Flickr30K | | | COCO | | |
| | R@1 | R@5 | R@10 | R@1 | R@5 | R@10 | R@1 | R@5 | R@10 | R@1 | R@5 | R@10 |
| *Contrastive Learning on image-text pairs* | | | | | | | | | | | | |
| CLIP `ViT-B` | 58.8 | 83.3 | 89.8 | 30.5 | 56.0 | 66.8 | 77.8 | 95.0 | 98.2 | 51.0 | 74.9 | 83.5 |
| BLIP `ViT-L` | 70.0 | 91.2 | 95.2 | 48.4 | 74.4 | 83.2 | 75.5 | 95.1 | 97.7 | 63.5 | 86.5 | 92.5 |
| CLIP `ViT-L` | 67.3 | 89.0 | 93.3 | 37.0 | 61.6 | 71.5 | 87.2 | 98.3 | 99.4 | 58.1 | 81.0 | 87.8 |
| EVA-02-CLIP `5B` | 78.8 | 94.2 | 96.8 | 51.1 | 75.0 | 82.7 | **93.9** | **99.4** | **99.8** | **68.8** | **87.8** | **92.8** |
| *Contrastive Learning only on text pairs* | | | | | | | | | | | | |
| E5-V | **79.5** | **95.0** | **97.6** | **52.0** | **76.5** | **84.7** | 88.2 | 98.7 | 99.4 | 62.0 | 83.6 | 89.7 |

Table 1: Zero-shot text-image retrieval performance on Flickr30K and COCO.

## 4.2 COMPOSED IMAGE RETRIEVAL

To understand the effectiveness of E5-V in representing interleaved visual and language inputs, we evaluate it on composed image retrieval tasks with two popular datasets: FashionIQ Wu et al. (2021) and CIRR Liu et al. (2021). This task focuses on retrieving images based on interleaved inputs, which requires the model to retrieve target images based on modified reference images, where the modification is described in the text. For FashionIQ, it contains three subtypes of fashion products: Dress, Shirt and Toptee. Given a picture of a fashion product and a modification corresponding to the style, the model needs to retrieve the target image that matches the modification. For CIRR, it extend FashionIQ on real-life images, which has more diverse images and modifications.

We compare E5-V with several zero-shot image-composed baselines: Pic2Word Saito et al. (2023), Context-I2W Tang et al. (2024), LinCIR Gu et al. (2024), the LLM-based method CIReVL Karthik et al. (2023), and the current state-of-the-art method iSEARLE-XL Agnolucci et al. (2024). For a fair comparison, we report the results of all baseline models using the large visual encoder CLIP `ViT-L`, as in E5-V. Note that the E5-V also freezes visual encoder same as other baselines. These baselines are designed exclusively for zero-shot composed image retrieval tasks and can not apply to other tasks. Most of the baselines are not end-to-end embedding interleaved inputs, which introduce complex pipelines like textual inversion. For example, CIReVL requires captioning an image first, generating the target image caption based on LLMs, and then retrieving the target image based on the caption. However, E5-V can directly represent the interleaved visual and language inputs with prompts without any textual inversion.

To represent the interleaved inputs for E5-V, we use the following prompts for FashionIQ and CIRR. For FashionIQ, which requires the model to mainly represent the style of the fashion product, we can directly let E5-V represent the style of the corresponding fashion products. Since the evaluation of FashionIQ is split into three subtypes, including Dress, Shirt, and Toptee, we can also provide the subtype information in the prompts. For CIRR, we can directly let E5-V modify the image based on the modification described in the text and then represent the modified image in one word. Although these prompts are unseen during training and have a complex format, E5-V can still correctly represent the interleaved inputs, even in specific domains like fashion products.

> **Composed image prompt for FashionIQ:**
> `<image>` `change the style of this shirt/dress/toptee to` `<text>`
> `Describe this modified shirt/dress/toptee in one word based on its`
> `style:`
>
> **Image prompt for FashionIQ:**
> `<image>`
> `Describe this shirt/dress/toptee in one word based on its style:`

```
Composed image prompt for CIRR:
<image> modify this image with <text>
Describe modified image in one word:
 Image prompt for CIRR:
<image>
Describe this image in one word:
```

We report the composed image retrieval performance of CIRR and FashionIQ on Table 2 and 3. All methods use CLIP `ViT-L` as the visual encoder. The results of other baselines are directly from their original papers. Following previous works, we report Recall@K for K=1, 5, 10, and 50 on CIRR test set with their test evaluation server, and report Recall@K for K=10, 50 on three subsets of FashionIQ. For the settings of E5-V, we use original E5-V without additional fine-tuning on specific datasets and tricks like textual inversion. E5-V directly represents the interleaved inputs and image inputs with above prompts and uses the last token embeddings to represent the multimodal embeddings.

| | Recall@K | | | |
|---|---|---|---|---|
| **Method** | K=1 | K=5 | K=10 | K=50 |
| Pic2Word | 23.90 | 51.70 | 65.30 | 87.80 |
| Context-I2W | 25.60 | 55.10 | 68.50 | 89.80 |
| LinCIR | 25.04 | 53.25 | 66.68 | – |
| CIReVL | 24.55 | 52.31 | 63.92 | 86.34 |
| iSEARLE-XL | 25.40 | 54.05 | 67.47 | 88.92 |
| **E5-V** | **33.90** | **64.12** | **75.88** | **93.54** |

Table 2: Zero-shot composed image retrieval performance on CIRR.

Compared to zero-shot composed image retrieval baselines, E5-V achieves significant improvements on both the CIRR and FashionIQ datasets without using techniques like textual inversion or annotation. Specifically, E5-V outperforms the current state-of-the-art method iSEARLE-XL by 8.50% on Recall@1 and 10.07% on Recall@5 on CIRR. For FashionIQ, E5-V outperforms by 2.56% on Recall@10 and 4.24% on Recall@50 compared to iSEARLE-XL, which demonstrates the great ability of E5-V understanding the interleaved visual and language inputs and representing them correctly.

| **Method** | Shirt | | Dress | | Toptee | | Average | |
|---|---|---|---|---|---|---|---|---|
| | R@10 | R@50 | R@10 | R@50 | R@10 | R@50 | R@10 | R@50 |
| Pic2Word | 26.20 | 43.60 | 20.00 | 40.20 | 27.90 | 47.40 | 24.70 | 43.70 |
| Context-I2W | 29.70 | 48.60 | 23.10 | 45.30 | 30.60 | 52.90 | 27.80 | 48.93 |
| LinCIR | 29.10 | 46.81 | 20.92 | 42.44 | 28.81 | 50.18 | 26.28 | 46.49 |
| CIReVL | 29.47 | 47.40 | **24.79** | 44.76 | 31.36 | 53.65 | 28.55 | 48.57 |
| iSEARLE-XL | 31.80 | 50.20 | 24.19 | 45.12 | 31.72 | 53.29 | 29.24 | 49.54 |
| **E5-V** | **36.36** | **56.43** | 23.75 | **47.45** | **35.29** | **57.47** | **31.80** | **53.78** |

Table 3: Zero-shot composed image retrieval performance on FashionIQ.

### 4.3 IMAGE-IMAGE RETRIEVAL

By unifying multimodal representations into the same embedding space with prompts, E5-V demonstrates a strong ability to understand text through visual input and represent it accurately. To validate this, we designed an image-image retrieval task based on Flickr30K and COCO, referred to as I2I-Flickr30K and I2I-COCO. We rendered all textual captions in the datasets as images and used the embeddings of these images as the caption embeddings. The detailed implementation of text rendering can be found in Appendix A. For the prompts of E5-V, we simply used the image prompt in text-image retrieval tasks to represent images.

We report the results of CLIP, BLIP, EVA-02-CLIP, and E5-V in Table 4. Compared to text-image retrieval tasks, we notice that the performance of baselines drops significantly on image-image retrieval tasks, which indicates the difficulty of understanding text through visual input and representing it accurately. Due to separate visual and language encoders, these models struggle

to understand the textual information via images by using their visual encoders. However, E5-V correctly represents text through visual input and shows outstanding results on these two datasets.

| Method | image retrieval | | | | | | text (render as image) retrieval | | | | | |
| | I2I-Flickr30K | | | I2I-COCO | | | I2I-Flickr30K | | | I2I-COCO | | |
| | R@1 | R@5 | R@10 | R@1 | R@5 | R@10 | R@1 | R@5 | R@10 | R@1 | R@5 | R@10 |
|---|---|---|---|---|---|---|---|---|---|---|---|---|
| BLIP ViT-L | 3.3 | 9.5 | 14.2 | 1.1 | 3.4 | 5.4 | 9.0 | 22.1 | 31.1 | 4.2 | 11.4 | 16.7 |
| CLIP ViT-L | 3.8 | 10.8 | 16.1 | 1.5 | 4.4 | 6.6 | 27.7 | 52.7 | 63.9 | 10.9 | 24.9 | 33.2 |
| EVA-02-CLIP 5B | 18.8 | 37.8 | 46.9 | 6.3 | 16.0 | 22.9 | 42.3 | 71.0 | 81.4 | 17.2 | 35.9 | 46.6 |
| E5-V | **67.8** | **89.2** | **93.6** | **41.2** | **66.7** | **76.2** | **79.5** | **95.2** | **97.8** | **51.6** | **76.8** | **84.9** |

Table 4: Zero-shot image-image retrieval performance on I2I-Flickr30K and I2I-COCO.

## 4.4 Sentence Embeddings

Since E5-V is trained on text pairs, it also shows strong performance in representing textual inputs. We evaluate E5-V on the sentence embedding tasks using 7 STS tasks. Compared to other sentence embedding methods, including SimCSE-RoBERTa Gao et al. (2021), PromptRoBERTa Jiang et al. (2022), and LLM-based methods such as SGPT Muennighoff (2022), ST5-Enc Ni et al. (2021), and PromptEOL Jiang et al. (2023), E5-V, as a universal multimodal model, achieves the best performance on the STS tasks in Table 5, demonstrating its strong ability to represent textual inputs according to their semantics.

| Method | STS12 | STS13 | STS14 | STS15 | STS16 | STS-B | SICK-R | Avg. |
|---|---|---|---|---|---|---|---|---|
| SimCSE-RoBERTa | 76.53 | 85.21 | 80.95 | 86.03 | 82.57 | 85.83 | 80.50 | 82.52 |
| PromptRoBERTa | 76.75 | 85.93 | 82.28 | 86.69 | 82.80 | 86.14 | 80.04 | 82.95 |
| SGPT | 74.28 | 85.35 | 79.21 | 85.52 | 82.54 | 85.50 | 79.53 | 81.70 |
| ST5-Enc | 80.10 | 88.75 | 84.70 | 88.86 | 85.17 | 86.77 | 80.39 | 84.96 |
| PromptEOL | 79.16 | 90.22 | 85.40 | 88.99 | 86.25 | 88.37 | 81.51 | 85.70 |
| E5-V | 80.03 | 89.94 | 85.67 | 89.09 | 85.89 | 87.88 | 83.51 | **86.00** |

Table 5: Sentence embeddings performance on STS tasks.

## 5 Discussion

### 5.1 Effect of the Representation Method

To validate the effectiveness of our prompt representation method, we compare it with two other methods: 1) Last: using the last token embeddings of the input as the multimodal embeddings, and 2) Prompt: using the same prompt as our methods, but removing *in one word:* in prompt. We report the performance of these methods with and without fine-tuning in Table 6. For the fine-tuning, we fine-tune each method with corresponding prompts on sentence pairs with contrastive learning following the same training settings as E5-V.

Our method shows significant improvements on all tasks compared to the Last and Prompt. For the setting without fine-tuning, we observe that our method can directly leverage the MLLM to represent the multimodal embeddings. However, other methods cannot represent the multimodal inputs properly. We also find that these methods have a large `modality gap` between image and text embeddings, as shown in Appendix B. For the setting with fine-tuning, we also observe the performance gap between our method and other methods. Although Prompt uses same template with our method and just removes *in one word:* in it, it still shows significant performance drop compared to our method especially on tasks with visual inputs. One possible reason may be the `modality gap` limit it to transfer the single modality representation capabilities learned on text inputs to multimodal embeddings.

| Method | Flickr30K | COCO | CIRR | FashionIQ | I2I-Flickr30K | I2I-COCO | STS. | Avg. |
|--------|-----------|------|------|-----------|---------------|----------|------|------|
| *Without fine-tuning* | | | | | | | | |
| Last | 8.9/4.1 | 4.6/3.3 | 7.4 | 3.4 | 3.0/5.1 | 0.6/1.8 | 58.5 | 9.2 |
| Prompt | 22.4/5.5 | 8.9/1.3 | 1.2 | 1.9 | 3.7/7.3 | 0.4/3.6 | 57.5 | 10.3 |
| Our | **82.8/90.4** | **60.3/67.4** | **38.4** | **32.4** | **67.0/75.4** | **41.8/49.3** | **75.8** | **61.9** |
| *With fine-tuning* | | | | | | | | |
| Last | 91.8/94.6 | 69.8/73.7 | 31.6 | 16.6 | 79.4/90.7 | 53.2/64.2 | 84.1 | 68.2 |
| Prompt | 93.5/96.6 | 74.6/77.0 | 62.3 | 32.0 | 85.4/92.7 | 62.4/70.5 | 85.1 | 75.6 |
| Our | **95.0/98.7** | **76.5/83.6** | **66.6** | **53.8** | **89.2/95.2** | **66.7/76.8** | **86.0** | **80.7** |

Table 6: Effect of the representation method on different tasks. We report Recall@50 for FashionIQ, Spearmans correlation for STS tasks and Recall@5 for other tasks. For CIRR, we report the results on the validation set.

## 5.2 EFFECT OF SINGLE MODALITY TRAINING

We also compare single modality training with multimodal training. For multimodal training, we train the MLLM on 558K text-image pairs from CC3M using the same training settings and prompts as single modality training. We report the performance of single modality training and multimodal training on different tasks in Table 7. We find that MLLM achieves better multimodal embeddings with single modality training. Even on the image-text retrieval tasks, where multimodal training uses similar training data, single modality training still shows better performance. For other tasks, we notice that multimodal training cannot represent the interleaved inputs in FashionIQ and CIRR, or text inputs in STS well, which leads to a performance drop compared to single modality training. Moreover, single modality training is more efficient by removing the visual encoder and only uses 32 max tokens for text inputs, significantly reducing the training time compared to multimodal training. Single modality training only takes 1.5 hours on 32 V100 GPUs, while multimodal training takes 34.9 hours under same environment.

| | Training time | Flickr30K | COCO | CIRR | FashionIQ | I2I-Flickr30K | I2I-COCO | STS. | Avg. |
|--|---------------|-----------|------|------|-----------|---------------|----------|------|------|
| Multimodal training | 34.9h | 93.5/97.8 | 76.0/83.1 | 35.5 | 30.8 | 84.2/93.0 | 64.1/73.4 | 72.7 | 73.1 |
| Single modality training | 1.5h | **95.0/98.7** | **76.5/83.6** | **66.6** | **53.8** | **89.2/95.2** | **66.7/76.8** | **86.0** | **80.7** |

Table 7: Effect of single modality training on different tasks. We measure the training time on 32 V100 GPUs.

## 5.3 ZERO-SHOT INSTRUCTION FOLLOWING ABILITY ON MULTIMODAL EMBEDDINGS

We find an interesting ability of E5-V to represent inputs based on fully zero-shot instructions. Although E5-V is trained on text inputs with the static prompt, it can correctly represent visual and interleaved inputs based on unseen prompts. These prompts can be more detailed and specific based on the tasks. For example, in FashionIQ, a specific domain dataset about fashion products, we can design specific prompts to let E5-V embed the image based on their styles. Moreover, the interaction between visual and language inputs in E5-V can also be more detailed, such as `change the style of this shirt to`. Compared to other methods, which simply fuse the visual and language inputs, E5-V provides a more nuanced and specific approach.

## 6 CONCLUSION

In this work, we propose E5-V, a MLLM based universal multimodal model that can represent interleaved visual and language inputs accurately. E5-V uses the prompt based representation method to unify multimodal representations into the same embedding space without additional fine-tuning or tricks. With single modality training, E5-V achieves strong performance on various tasks, including text-image retrieval, composed image retrieval, image-image retrieval, and sentence embeddings. We also conduct extensive ablation studies to validate the effectiveness of our method.

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

## A    IMPLEMENTATION OF TEXT RENDERING IN IMAGE-IMAGE RETRIEVAL

We introduce the implementation of text rendering in image-image retrieval tasks. We use the `PIL` library to render the text as an image by using the `Arial` font with 40 pixels font size and 800×400 resolution. To fit the long text, we also automatically break the text into multiple lines to fit the image size. We provide a example of rendering text as image in Figure 4.

Figure 4: An example of rendering text as image with text "A man in a black shirt rides an elephant as a man walks near it down a street."

The python code is shown below:

```python
from PIL import Image, ImageDraw, ImageFont
def create_text_image(text):
    image_width=800
    image_height=400
    font_path="arial.ttf"
    font_size=40
    background_color=(255, 255, 255)
    text_color=(0, 0, 0)

    image = Image.new('RGB', (image_width, image_height), color=
        background_color)
    draw = ImageDraw.Draw(image)
    font = ImageFont.truetype(font_path, font_size)

    # padding
    max_text_width = image_width - 40

    # Break line based on length
    lines = []
    words = text.split()
    while words:
        line = ''
        while words and draw.textlength(line + words[0], font=font) <=
            max_text_width:
            line += (words.pop(0) + ' ')
        lines.append(line)

    # Calculate the position for the text
    total_text_height = sum(draw.textbbox((0, 0), line, font=font)[3] -
        draw.textbbox((0, 0), line, font=font)[1] for line in lines)
    text_x = 20
    text_y = (image_height - total_text_height) // 2
```

```
# Add text to image
for line in lines:
    draw.text((text_x, text_y), line, font=font, fill=text_color)
    text_y += draw.textbbox((0, 0), line, font=font)[3] - draw.
        textbbox((0, 0), line, font=font)[1]

return image
```

## B    DISTRUIBUTION OF IMAGE AND TEXT EMBEDDINGS WITH DIFFERENT REPRESENTATION METHODS

We visualize the distribution of multimodal embeddings from MLLM with three different representation methods: Last, Prompt, and Our. The embeddings are directly from LLaVA-NeXT-8B without fine-tuning on any specific dataset. Our method removes `modality gap` between image and text embeddings, which is shown in Figure 5. The prompt for each method is following:

---

**Text prompt in Last:**
`<text>`
**Image prompt in Last:**
`<image>`

**Text prompt in Prompt:**
`<text>`
`Summary above sentence:`
**Image prompt in Prompt:**
`<image>`
`Summary above image:`

**Text prompt in Our:**
`<text>`
`Summary above sentence in on word:`
**Image prompt in Our:**
`<image>`
`Summary above image in one word:`

---

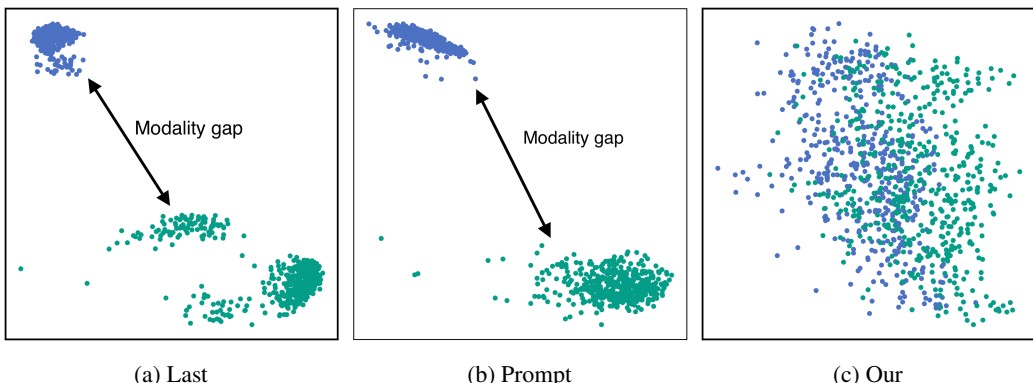

(a) Last                    (b) Prompt                    (c) Our

Figure 5: Distribution of image embeddings and text embeddings with different representation methods.

## C    MULTIMODAL EMBEDDINGS WITH DIFFERENT MLLMS

We also evaluate the performance of E5-V with different MLLMs, including Phi3, LLaVA 1.6, and LLaVA-NeXT in Table 8. Despite the different LLM in each MLLM, E5-V also show strong performance with same prompts. Among these MLLMs, LLaVA-NeXT shows the best performance with fine-tuning benfit from the high capacity of LLM.

| Method | Flickr30K | COCO | CIRR | FashionIQ | I2I-Flickr30K | I2I-COCO | STS. | Avg. |
|---|---|---|---|---|---|---|---|---|
| *Without fine-tuning* | | | | | | | | |
| Phi3 | 80.0/89.9 | 55.3/70.3 | 43.7 | 31.5 | 71.7/83.8 | 46.3/61.3 | 72.1 | 64.2 |
| LLaVA 1.6 (Mistral) | 80.5/89.8 | 59.1/70.8 | 28.3 | 33.4 | 53.8/78.8 | 41.8/55.4 | 73.5 | 60.5 |
| LLaVA-NeXT (LLaMA 3) | 82.8/90.4 | 60.3/67.4 | 38.4 | 32.4 | 67.0/75.4 | 41.8/49.3 | 75.8 | 61.9 |
| *With fine-tuning* | | | | | | | | |
| Phi3 | 93.0/96.9 | 71.5/81.1 | 58.4 | 48.1 | 89.3/95.5 | 65.1/77.1 | 85.2 | 78.3 |
| LLaVA 1.6 (Mistral) | 94.6/97.4 | 74.9/83.1 | 68.9 | 50.1 | 86.5/93.0 | 65.9/73.4 | 84.9 | 79.3 |
| LLaVA-NeXT (LLaMA 3) | 95.0/98.7 | 76.5/83.6 | 66.6 | 53.8 | 89.2/95.2 | 66.7/76.8 | 86.0 | 80.7 |

Table 8: Performance of E5-V with different MLLMs on different tasks.

