# OpenReview forum: "E5-V: Universal Embeddings with Multimodal Large Language Models"
_ICLR.cc/2025/Conference — Submitted to ICLR 2025_

### Official Review · Reviewer_7pfu · 2024-10-31

**Soundness:** 3
**Presentation:** 2
**Contribution:** 3
**Rating:** 5
**Confidence:** 4

**Summary:**

This paper focuses on obtaining representations of texts and images
from multimodal LLMs (MLLMs).  Authors approach this goal by proposing
E5-V, which leverages prompts like "summarize above image in one word"
along with an image
to input to the MLLMs to close the modality gap.  E5-V is a Llava-based
MLLM finetuned on text-only data (NLI) to extract multimodal
representations. It also supports for embedding interleaving text and
image sequences, which potentially supports for new applications.

Authors show that E5-V outperforms CLIP-based methods in text-to-image
retrieval benchmarks. Usage of E5-V on composed image retrieval and
image-to-image retrieval is also presented.

**Strengths:**

- E5-V is simple, intuitive, and can be potentially adopted in various applications.

- Training E5-V seems lightweight -- it just requires training on text-only data.

- E5-V highlights the potential of extracting representations from LLM-based models. Expanding the scope of the current research and
  real-world applications where CLIP-like models are still extensively used.

- Representing modality-interleaved sequences is beneficial and can unlock many new applications.

**Weaknesses:**

- Compared with CLIP-like models, inference with LLM-based E5-V seems
  still heavy (e.g., 8B of E5-V vs 5B of EVA-CLIP).

- More in-depth analyses are desirable/needed to show the inner
  workings of E5-V, e.g., how did it close the modality gap, by just
  using a prompt? Also, some qualitative examples are necessary. E.g.,
  is E5-V better at encoding higher-level semantic information than
  some low-level features like textures?

- Image-to-text retrieval performance (Tab1) on standard benchmarks
  Flickr30K and COCO are not good and consistently worse than
  baselines. This is counter-intuitive, given that E5-V was finetuned
  on the  NLI datasets, which is text data.

So overall I feel the paper is a bit pre-mature, and more rounds would
be beneficial.

**Questions:**

In Tab1, why E5-V achieved worse text-retrieval results, consistently,
than EVA-02-CLIP-5B? E5-V was finetuned on NLI datasets, which are
supposed to improve its ability of embedding texts. But this seems not
the case. What are the hypotheses of this observation?

---

> ### Author Response · Authors · 2024-11-19
> **Response to Reviewer 7pfu**
>
> We sincerely thank you for your helpful feedback and insightful comments. We address your comments and questions below.
>
> > **W1** Compared with CLIP-like models, inference with LLM-based E5-V seems still heavy (e.g., 8B of E5-V vs 5B of EVA-CLIP).
>
> We have reported the results of E5-V on smaller MLLMs like phi-3, which has only 4 billion total parameters, in Appendix C. It achieves an average score of 78.3 across seven tasks, compared to 80.7 for E5-V, with only half of the parameters.
> Furthermore, we believe that with the development of MLLMs, it will be possible to further reduce model size while maintaining performance.
>
>
> > **W2** More in-depth analyses are desirable/needed to show the inner workings of E5-V, e.g., how did it close the modality gap, by just using a prompt? Also, some qualitative examples are necessary. E.g., is E5-V better at encoding higher-level semantic information than some low-level features like textures?
>
> Thank you for your suggestion. We will add more in-depth analyses to show the inner workings of E5-V.
>
> For the modality gap, we find that simply using our prompt-based representation method like "Summary in one word" can effectively closing the modality gap.
> We also provide a quantitative metric using $\|\|\vec{\Delta}\_{\text{gap}}\|\|$ from [1]. The table below shows the results without and with our prompt-based representation method. Our prompt-based representation method can significantly reduce the modality gap, as the $\|\|\vec{\Delta}\_{\text{gap}}\|\|$ has dropped from 0.6655 to 0.2893.
>
> |     | $\|\|\vec{\Delta}_{\text{gap}}\|\|$  |
> |-------------------|------------------------------|
> | w/o our method | 0.6655                       |
> | w/ our method     | 0.2893                       |
>
>
>
> > **W3\&Q1** Image-to-text retrieval performance (Tab1) on standard benchmarks Flickr30K and COCO are not good and consistently worse than baselines. This is counter-intuitive, given that E5-V was finetuned on the NLI datasets, which is text data.
> In Tab1, why E5-V achieved worse text-retrieval results, consistently, than EVA-02-CLIP-5B? E5-V was finetuned on NLI datasets, which are supposed to improve its ability of embedding texts. But this seems not the case. What are the hypotheses of this observation?
>
>
> First, E5-V is trained only on text pairs, with no images used during contrastive learning. And the image embeddings of E5-V are not tuned during training. In contrast, CLIP is trained on both text and image embeddings using large-scale image-text pairs, which naturally gives it an advantage in image-text retrieval tasks. For text retrieval in Table 1, text embeddings are used to retrieve from multiple image embeddings, which requires not only high text embeddings but also effective image embeddings. In terms of text representation, we also evaluated our method on sentence embeddings and found that it outperforms the previous state-of-the-art in Table 5.
>
> Second, EVA-CLIP highlights the importance of visual encoder size[2]. It achieves strong performance by scaling its visual encoder to 4.8B parameters. However, the visual encoder of E5-V has only 428M parameters and remains frozen during both fine-tuning and MLLM pretraining. Compared to CLIP ViT-L, which uses the same visual encoder with E5-V, E5-V demonstrates significant improvements in Table 1.
>
>
>
> [1] Liang V W, Zhang Y, Kwon Y, et al. Mind the gap: Understanding the modality gap in multi-modal contrastive representation learning[J]. NIPS, 2022, 35: 17612-17625.
>
> [2] Sun Q, Fang Y, Wu L, et al. Eva-clip: Improved training techniques for clip at scale[J]. arXiv preprint arXiv:2303.15389, 2023.

---

> > ### Comment · Reviewer_7pfu · 2024-11-21
> > **Response to Authors**
> >
> > Thank you for replying to my questions.
> >
> > In general, I like the idea and the model, but still I am not completely clear about the text-retrieval performance.
> >
> > According to EVA-CLIP paper Tab4 (https://arxiv.org/pdf/2303.15389), EVA-02-CLIP-L/14+ outperforms E5-V as well, but with a smaller model size. So I am not completely convinced by the argument of model size.
> >
> > I understand that E5-V is just trained on some NLI dataset and already well-performing. However, the inferior text-retrieval performance discourages me to update my rating, if the claim is to improving over traditional multimodal training on image-text pairs (as in the abstract).

---

> > > ### Author Response · Authors · 2024-11-21
> > > **Response to Reviewer 7pfu**
> > >
> > > Thanks for your replying and acknowledging our work.
> > >
> > > But there seems to be some confusion regarding the claim in our abstract: `We propose a single modality training approach for E5-V, where the model is trained exclusively on text pairs. This method demonstrates significant improvements over traditional multimodal training on image-text pairs, while reducing training costs by approximately 95%. ` The `traditional multimodal training` is refer to traditional multimodal training on the MLLM compared to single modality training.
> > > This claim is supported by Section 5.2,  we have compared our single modality training with traditional multimodal training on MLLM. Our training method outperform traditional multimodal training with less training costs.
> > >
> > > We acknowledge that the current E5-V underperforms compared to strong CLIP models when evaluated on text retrieval performance in text-image retrieval.  We think it may be attributed to the visual encoder used in E5-V and the training data.
> > > The visual capability of E5-V is still based on an untuned CLIP ViT-L, as shown in the results in Table 1.
> > > We would also like to point out that E5-V is designed for universal multimodal embeddings. It demonstrates strong performance on other tasks, such as composed-image retrieval, image-image retrieval, and sentence embeddings, where CLIP-like methods fall short.

---

### Official Review · Reviewer_CUEB · 2024-11-03

**Soundness:** 3
**Presentation:** 3
**Contribution:** 3
**Rating:** 6
**Confidence:** 3

**Summary:**

E5-V is a framework that adapts multimodal large language models (MLLMs) for universal multimodal embeddings. By training solely on text pairs, E5-V reduces training costs by 95% and demonstrates high effectiveness across tasks, bridging modality gaps efficiently.

**Strengths:**

Using MLLM so that E5-V could deal with interleaved input and using small batch size.

Only trained on text-only data leads to better performance compared to the model trained on multimodal data.

Use “summary XXX in on word” to bridge the modality gap between text and vision effectively

**Weaknesses:**

The "summary XXX in one word" prompt enables the model to handle simple image or text inputs. If the input complexity increases, a word cannot illustrate the input text /image well.

For new multi-model task, the new prompt needs to be designed to ensure effective performance.

**Questions:**

Since there is no training for modality encoder and projector, how can the modality encoder and projector generalize to the model in the
inference time?

E5-V is trained on text-only data, yet it shows minimal improvement in text-only tasks (e.g., sentence embeddings: 86.00 vs. previous SOTA 85.70). However, it achieves significant gains in image-related tasks, such as image retrieval. Why does this difference in improvement occur?

---

> ### Author Response · Authors · 2024-11-19
> **Response to Reviewer CUEB**
>
> We sincerely thank you for your helpful feedback and insightful comments. We address your comments and questions below.
>
> > **W1** The "summary XXX in one word" prompt enables the model to handle simple image or text inputs. If the input complexity increases, a word cannot illustrate the input text /image well.
>
> The design of the prompt does not compress the input directly into a single word. Instead, we use the last hidden states of the MLLM as the embedding.
> Due to the next token prediction in MLLM, this embedding reflects the probability distribution of words corresponding to the input, giving related words a higher probability.
>
> We also test E5-V on complex tasks like CIRR in Table 2. This task involves queries that contain both text and images, such as changing an image of a Cavalier King Charles Spaniel (a breed of dog) with the text "Be a same breed dog with his puppy running." E5-V accurately represents inputs like these and outperforms the previous state-of-the-art by 8.5% on Recall@1.
>
> > **W2** For new multi-model task, the new prompt needs to be designed to ensure effective performance.
>
> Compared to previous methods, E5-V does not require fine-tuning for new multi-modal tasks. And new prompts are only used for composed image retrieval task in our paper, which involve interleaved inputs or specific domain.
> Additionally, we believe that modifying prompts is easier than fine-tuning or designing a new model for a specific task. We also observed that E5-V demonstrates a strong ability to follow instructions for multimodal embeddings, as discussed in Section 5.3.
>
> > **Q1** Since there is no training for modality encoder and projector, how can the modality encoder and projector generalize to the model in the inference time?
>
> We think it benefits from the proposed representation method, which removes the gap between different modalities, as shown in Figure 3. Since different modalities share the same embedding space, the learned representation can easily transfer to the modality encoder and projector. We report an ablation study on this in Table 6 and Figure 5. Besides, we release the code, including the training pipeline, in the supplementary material.
>
> > **Q2** E5-V is trained on text-only data, yet it shows minimal improvement in text-only tasks (e.g., sentence embeddings: 86.00 vs. previous SOTA 85.70). However, it achieves significant gains in image-related tasks, such as image retrieval. Why does this difference in improvement occur?
>
> E5-V focuses on bridging the text representation ability to other modalities by single. We demonstrate a strong improvement by transferring text representation ability to other modalities. However, when it comes to sentence embeddings, this task only tests text representation, resulting in minimal improvement compared to previous LLM-based methods.

---

### Official Review · Reviewer_S4th · 2024-11-03

**Soundness:** 2
**Presentation:** 3
**Contribution:** 3
**Rating:** 6
**Confidence:** 3

**Summary:**

This paper presents a method of training a multimodal model that can output multimodal representations. In contrast to conventional multimodal models such as CLIP, which consists of separate encoders for different modalities, the proposed E5-V is based on a single pretrained MLLM (multimodal large language model) such as LLaVA-Next. E5-V is trained on only text data (natural language inference dataset) and requires a lower computational cost than the previous models by leveraging the knowledge of the MLLM. The authors conducted experiments on text-image retrieval, composed image retrieval, image-image retrieval, and sentence embedding tasks. The experiments demonstrated that the proposed training method works well.

**Strengths:**

1. The paper is well written. I could understand their motivation, the proposed method, and the experimental results.
2. The proposed method is simple but effective. If we have a good pretrained MLLM, we can easily fine-tune it for multimodal representations. This method can be applied to other combinations of two or more modalities as well.
3. A trained embedding space is free from the modality gap issue, one of the important issues of contrastive learning-based multimodal models.
4. The authors conducted several experiments to show the proposed training method works well.

**Weaknesses:**

- I acknowledge that the proposed training method is efficient if we have a good enough pretrained MLLM and that a trained embedding space is free from the modality gap. These properties are excellent. However, I doubt if the comparative experiments are fair in terms of model size. The model size of E5-V (LLaVA-Next-8B) is 8B, which is larger than CLIP, BLIP, and even EVA-CLIP used for comparison. As shown in the [EVA-CLIP paper](https://arxiv.org/abs/2303.15389), the model size affects its performance. If possible, a smaller size of E5-V should be evaluated/compared as well. I know that LLMs show emergent abilities and smaller models might have no/weak capabilities of encoding images/text very well. Even if so, showing comparisons will be informative to readers.

**Questions:**

- I would appreciate the authors' response about my comment in "Weaknesses".
- I recommend placing some samples of query and retrieved image pairs from the composed image retrieval and image-image retrieval tasks in the main body and/or Appendix. Such visualization will provide readers with a clear vision of the differences between the conventional and proposed models.

---

> ### Author Response · Authors · 2024-11-19
> **Response to Reviewer S4th**
>
> We sincerely thank you for your helpful feedback and insightful comments. We address your comments and questions below.
>
> > **W1**  I acknowledge that the proposed training method is efficient if we have a good enough pretrained MLLM and that a trained embedding space is free from the modality gap. These properties are excellent. However, I doubt if the comparative experiments are fair in terms of model size. The model size of E5-V (LLaVA-Next-8B) is 8B, which is larger than CLIP, BLIP, and even EVA-CLIP used for comparison. As shown in the EVA-CLIP paper, the model size affects its performance. If possible, a smaller size of E5-V should be evaluated/compared as well. I know that LLMs show emergent abilities and smaller models might have no/weak capabilities of encoding images/text very well. Even if so, showing comparisons will be informative to readers.
>
> Thank you for acknowledging our work. We respond with the following three points:
>
> 1. **Model Size**:  EVA-CLIP highlights the importance of visual encoder size[2]. It achieves strong performance by scaling its visual encoder from 86M to 4.8B parameters. Despite the fact that the visual encoder of E5-V is only a 428M CLIP, E5-V outperforms the 4.4B visual encoder of EVA-CLIP in image retrieval. Furthermore, the visual encoder of E5-V remains frozen during both fine-tuning and pretraining. Compared to CLIP ViT-L, which uses the same visual encoder, E5-V demonstrates significant improvements, as shown in Table 1.
>
> 2. **Training Approach**: E5-V is trained using contrastive learning with a limited number of text pairs, and its image embeddings are not tuned during training. In contrast, EVA-CLIP is trained on large-scale image-text pairs with contrastive learning, which naturally gives it an advantage in image-text retrieval tasks.
>
> 3. **Performance on Smaller Models**: We also report performance of our method on a smaller MLLM (Phi-3 with 4B total parameters) in Appendix C. It achieves an average score of 78.3 across seven tasks, compared to 80.7 for E5-V, while using only half the parameters.
>
> > **Q1** I recommend placing some samples of query and retrieved image pairs from the composed image retrieval and image-image retrieval tasks in the main body and/or Appendix. Such visualization will provide readers with a clear vision of the differences between the conventional and proposed models.
>
> Thank you for your suggestions. We will add examples of the corresponding tasks to help readers better understand them and incorporate the polishing advices in the next version.
>
> [1] Sun Q, Fang Y, Wu L, et al. Eva-clip: Improved training techniques for clip at scale[J]. arXiv preprint arXiv:2303.15389, 2023.

---

> ### Comment · Reviewer_S4th · 2024-11-19
> **Response to Authors**
>
> Thank you very much for addressing my comments.
>
> I understand that training is not so heavy for the model size and that utilizing the knowledge of a pretrained MLLM is effective/efficient, but I am still not convinced about the fairness regarding the model size. ~When we compare `EVA-01-CLIP-g/14` and `EVA-01-CLIP-g/14+` (Table 1 of the EVA-CLIP paper), we can find out that the size of the text encoder affects the performance in the tested tasks.~ In addition, the model size will affect the computational cost or time required in inference as Reviewer 7pfu pointed out.
>
> As I said, I already acknowledge the novelty and effectiveness of the proposed approach. However, in my opinion, including both advantages and disadvantages/limitations is more honest and informative to readers than explaining only advantages, if there are some limitations. Let me keep the rating for now.

---

> ### Author Response · Authors · 2024-11-19
> **Response to Reviewer S4th**
>
> Thank you for your quick reply.  We also acknowledge that current inference of E5-V is still heavy compared to the CLIP.
>
> But we would like to point out that the improvement of  `EVA-01-CLIP-g/14+` compared to  `EVA-01-CLIP-g/14`  seems to be mainly due to different training data. The training data of `EVA-01-CLIP-g/14` is `LAION-400M` (400M image-text pairs), but the `EVA-01-CLIP-g/14+` is `Merged-2B` (2B image-text pairs).
>
>  As the Table 5 of EVA-CLIP paper, `Merged-2B` achieves the same performance as `LAION-400M` with only half of the training samples. They also mentioned `Only half samples were required to achieve the same top-1 accuracy when using the merged-2B dataset. It demonstrates the importance of dataset sizes and the significant convergence speed through merging the two datasets.`

---

> > ### Comment · Reviewer_S4th · 2024-11-19
> > **Response to Authors**
> >
> > Thank you for your kind reply. My analysis was wrong. I modified my previous post.
> >
> > Let me monitor the discussions with the other reviewers for a while. Thank you.

---

### Official Review · Reviewer_Rxx1 · 2024-11-04

**Soundness:** 2
**Presentation:** 3
**Contribution:** 3
**Rating:** 6
**Confidence:** 3

**Summary:**

The paper introduces E5-V, a prompt-based representation designed to achieve universal multimodal embeddings based on Multimodal Large Language Models (MLLMs). E5-V applies a cost-effective single-modality training approach using text-only data, and achieves better multimodal embeddings than image-text pairs across four tasks.

**Strengths:**

1. A new multimodal representation framework leveraging the multimodal understanding capability of MLLMs, with a clear motivation and a simple yet effective method.
2. Strong performance on text-image retrieval and composed image retrieval tasks with text-only training data, eliminating the need for costly multimodal training data and computing.

**Weaknesses:**

There are no major weaknesses. Just some minor weaknesses and a few questions in need of clarification.
Minor weaknesses:
1. The authors provide a visualization to illustrate the "modality gap". It would be better to provide a quantitative metric ( such as
https: //openreview. net/pdf? id=S7Evzt9uit3 ) to evaluate the gap between image embeddings and text embeddings from MLLMs w/o prompt-based representation.
2. Could you clarify whether the visualization of Figure 3(b) is before or after single modality training? Could you provide a visualization comparison of before and after single modality training?
3. One-word embedding might be not enough. I am wondering whether expanding the length of output words will lead to performance gains. For example, What if in Figure 2, there are multiple objects in the image and the instruction is to modify one object of them, will E5-V get it right?

**Questions:**

1. It is intriguing to see adding one modality could improve the performance of multiple modalities. I am wondering about the scalability of this method. What will happen if there is more text data included in fine-tuning? (Furthermore, will adding multimodal pairs further improve the performance?)
2. Some claims need supporting citations. For example, line 42-43, "adapting CLIP to universal multimodal embeddings will have shortcomings such as poor language understanding, limited real-world knowledge, .."
3. Regarding the robustness of this method, Would the difference in prompt during inference give substantial performance variations?

---

> ### Author Response · Authors · 2024-11-19
> **Response to Reviewer Rxx1**
>
> We sincerely thank you for your helpful feedback and insightful comments. We address your comments and questions below.
>
> > **W1** The authors provide a visualization to illustrate the "modality gap". It would be better to provide a quantitative metric....
>
> Thank you for your advice. We assess the gap using the quantitative metric $\|\|\vec{\Delta}\_{\text{gap}}\|\|$ from [1]. Below are the results without and with our prompt-based representation method, corresponding to the visualization in Figure 3. Our prompt-based representation method can significantly reduce the modality gap, as the $\|\|\vec{\Delta}\_{\text{gap}}\|\|$ has dropped from 0.6655 to 0.2893.
>
>
> |     | $\|\|\vec{\Delta}_{\text{gap}}\|\|$  |
> |-------------------|------------------------------|
> | w/o our method | 0.6655                       |
> | w/ our method     | 0.2893                       |
>
>
>
>
> > **W2** Could you clarify whether the visualization of Figure 3(b) is before or after single modality training? Could you provide a visualization comparison of before and after single modality training?
>
> Sorry for the confusion. Figure 3 shows the data before training. Due to the OpenReview format limitations, we provide the quantitative metric here by using $\|\|\vec{\Delta}_{\text{gap}}\|\|$:
>
> |     | $\|\|\vec{\Delta}_{\text{gap}}\|\|$  |
> |-------------------|------------------------------|
> |before training | 0.2893                       |
> |after training| 0.2189|
>
> We find that even though the single modality training, the modality gap is still reduced after training. We will add the quantitative metric and visualization comparison to the paper.
>
> > **W3** One-word embedding might be not enough. I am wondering whether expanding the length of output words will lead to performance gains. For example, What if in Figure 2, there are multiple objects in the image and the instruction is to modify one object of them, will E5-V get it right?
>
>
> We also tried a similar method, but it showed worse performance than E5-V. One possible reason is that increasing the length makes the model more inclined to use full sentences instead of single words. However, the one-word embedding, which is the last hidden state of the MLLM, doesn't correspond to just one word. Instead, it can reflect the probability distribution of words related to the inputs.
>
> For complex instructions, such as modifying an object, we also tested E5-V on similar tasks like CIRR in Table 2. This task includes queries like changing an image of a Cavalier King Charles Spaniel (a breed of dog) with "Be a same breed dog with his puppy running." E5-V accurately represents inputs like these and outperforms the previous state-of-the-art by 8.5% on Recall@1.
>
> > **Q1** It is intriguing to see adding one modality could improve the performance of multiple modalities. I am wondering about the scalability of this method. What will happen if there is more text data included in fine-tuning? (Furthermore, will adding multimodal pairs further improve the performance?)
>
> Thanks for your suggestion. The current training data for E5-V is only 273k sentence pairs. We are also interested in training E5-V on large-scale data that includes more text data or a mix of multimodal pairs. However, due to limitations in our current computational resources, we are unable to train on large-scale data. We have uploaded our training code in the supplementary material and will attempt this if computational resources become available.
>
> > **Q2** Some claims need supporting citations. For example, line 42-43, "adapting CLIP to universal multimodal embeddings will have shortcomings such as poor language understanding, limited real-world knowledge, .."
>
> This claim can be supported by [2], which found that the CLIP text encoder performs poorly in language understanding tasks like GLUE compared to BERT. We will add this citation to our paper.
>
> > **Q3**  Regarding the robustness of this method, Would the difference in prompt during inference give substantial performance variations?
>
> As we discuss in Section 5.3, we found that E5-V has a strong instruction-following ability for multimodal embeddings. Although it is trained on static text prompts, it can accurately represent visual and interleaved inputs with unseen prompts.
>  For example, if we change the prompt from:
> "<sent>\nSummary above sentence in one word:" and "<image>\nSummary above image in one word:"
> to
> "<sent>\nRepresent the text into single word:" and "<image>\nRepresent the image into single word:"
> It still achieves 95.0/98.4 compared to 95.0/98.7 with original prompt on flickr30k.
>
> [1] Liang V W, Zhang Y, Kwon Y, et al. Mind the gap: Understanding the modality gap in multi-modal contrastive representation learning[J]. NIPS, 2022, 35: 17612-17625.
>
> [2] Chen Z, Chen G, Diao S, et al. On the Difference of BERT-style and CLIP-style Text Encoders[C].Findings of ACL , 2023: 13710-13721.

---

> ### Comment · Reviewer_Rxx1 · 2024-11-26
>
> Thanks for the authors' detailed response. I think my most concerns have been addressed. Since my initial rating has reflected the core value of the work, I will keep my rating unchanged.

---

### Meta-Review · Area_Chair_2JZc · 2024-12-19

**Metareview:**

This paper presents E5-V, which turns a multimodal large language models to output multimodal representations. The idea of employing single-modality training approach using text-only data achieves better multimodal embeddings is novel.

The reviewers agree the framework is interesting and the idea of training on text-only data leads to a simply, intuitive, and lightweight tuning approach. The major concern from the reviewers is its lack of sufficient analysis and comparison to CLIP like models. Quota as “The idea of the paper is inspirational but the analysis is a bit less comprehensive.”

After a careful discussion, the reviewers agree this paper is promising on its direction, but still pre-mature.

**Additional Comments On Reviewer Discussion:**

The major concern is its lacking of sufficient analysis and comparison to CLIP like models. 3 of the 4 reviewers agrees on it and the concerns still exists after the discussion period.

---

### Decision · Program_Chairs · 2025-01-22

Reject